# Effects of Diamond Steps Exercises on Balance Improvement in Healthy Young and Older Adults: A Protocol Proposal [note 1]

**DOI:** 10.3390/healthcare11131834

**Published:** 2023-06-23

**Authors:** Shuangyan Shao, Tsubasa Mitsutake, Hitoshi Maruyama

**Affiliations:** 1Graduate School of Physical Therapy, International University of Health and Welfare, 2600-1 Kitakanemaru, Otawara 324-8501, Tochigi, Japan; hmaru@iuhw.ac.jp; 2Department of Physical Therapy, Faculty of Medicine, Fukuoka International University of Health and Welfare, 3-6-40 Momochihama, Sawara-ku, Fukuoka 814-0001, Fukuoka, Japan; mitutuba1012@gmail.com

**Keywords:** postural control, diagonal steps, subjective evaluation, difficulty, achievement, lightness/enjoyment, fall prevention, sprains, easy to sustain

## Abstract

Diamond step (DS) exercises are associated with multiple components of postural control and, thus, have the potential to efficiently improve balance ability. This study aimed to verify whether DS exercises contribute to improving balance ability. This study included 35 healthy young people and 29 older adults. DS exercises were performed continuously for 3 min, four times a week, for 1 month. Balance ability was assessed at baseline and after 1 and 2 months; eight items in total were examined: 30 s chair stand test, functional reach test, standing on one leg with eyes closed, time required for five rounds of DS, left–right DS, Y balance test, open–close stepping test, and finger-to-floor distance. The difficulty, achievement, and lightness/enjoyment of DS exercises were measured after the first practice and 1 month after beginning the exercises as subjective evaluations. Older adults showed improvement in seven of the eight items, with the exception being the one-legged stance with closed eyes. The subjective evaluation showed a decrease in the level of difficulty of DS exercises for older adults. DS exercises may improve balance by effectively utilizing various postural control strategies. These exercises can be effective and easy to implement, given their moderate difficulty level and self-efficacy.

## 1. Introduction

Balance ability is one of the most important physical abilities for daily life and sports activity. Older adults are generally prone to age-related decline in physical function, including muscle strength, flexibility, agility, and endurance [1]. Decline in physical function leads to impaired balance due to decreased physical activity, which can cause falls and bone fractures [2,3] and may even lead to frailty in some cases [4]. Frailty leads to decreased muscle mass, increased muscle fatigue, and reduced motor nerve function and transmission speed, a factor that influences falls, in older adults [5]. Older adults need regular exercise to live well and reduce the risk of falls [6]. Physical therapists should inform older adults about the positive effects of exercise on physical function and encourage them to engage in daily exercise [7]. Fall prevention involves a variety of interventions to improve physical and balance abilities. In healthy older adults, many balance practice models (with varying practice periods, frequency, and load), such as muscle strengthening by stair climbing [8], vibration [9], dance [10], balance practice using external disturbances [11], aerobic exercises, and double tasks [12], have been effective in improving balance. However, most methods require high intensity, long duration, or specific equipment, increasing the difficulty of practical application for older adults. Furthermore, exercise to improve physical function and balance in older adults is controversial in terms of the duration, type, intensity, and application of exercise [13].

On the other hand, sprains and chronic ankle instability (CAI) occur more frequently in young people during sports activities. Sprains occur in 75.1% of athletes during the men’s soccer season [14], and 50% of sprains in the basketball population result in CAI [15]. Although ankle sprains are often under-emphasized, as they may be considered as minor injuries, they have the highest recurrence rate, and approximately 70% of those who have sprained ankle joints are left with residual symptoms such as CAI, pain, swelling, and knee injury [16,17]. Balance exercises using a balance board could reduce the incidence of CAI [18]. Balance training has been reported to be effective in improving dynamic postural stability in CAI [19]. It has also been noted that postural control should be considered in the rehabilitation of CAI [20].

Therefore, both young and older adults require exercise and training to improve their physical and balance abilities, and training methods should be considered in both groups. Horak classifies the components of balancing capacity into the following categories: (1) static stability, (2) underlying motor systems, (3) functional stability limits, (4) verticality, (5) reactive postural control, (6) anticipatory postural control, (7) dynamic stability, (8) sensory integration, and (9) cognitive influences [21]. The Diamond Steps (DS) Test, which mainly comprises diagonal walking, includes seven of these balance elements, excluding verticality and cognitive function [22]. Training is designed to effectively improve physical function within a limited time frame. We hypothesize that practicing the DS used in the DS Test (DS exercises), which includes many balance elements, will effectively enhance multiple balance elements in a short time period.

The purpose of this study was to verify whether DS exercises contribute to an improvement in balance ability in young and older adults. Questionnaire interviews were also conducted to assess whether the DS exercises were easily sustainable for participants.

## 2. Materials and Methods

### 2.1. Participants

In total, 38 young adults (age: 27.8 ± 4.1 years, height: 164.5 ±10.4 cm, weight: 59.5 ± 13.1 kg, body mass index: 21.7 ± 2.8, 21 females) and 41 older adults aged >65 years (age: 73.0 ± 5.3 years, height: 154.5 ± 6.5 cm, weight: 57.2 ± 9.5 kg, body mass index: 23.9 ± 3.3, 36 females) living in the community were included in this study. Participants were excluded if they had conditions that would affect the intervention or assessment, such as (1) central nervous system disease; (2) joint disease or lower extremity pain; (3) comprehension or communication difficulty, e.g., dementia; or (4) vision problems. For recruitment, an advertisement was prepared and addressed to young and older adults residing in City A, and all applications were accepted on a voluntary basis. The advertisements clearly stated the age of the target participants, the nature of the cooperation, the physical requirement, and the fact that they could participate freely and withdraw at any time. Furthermore, the side effects were described as fatigue during the practice period and the risk of falling when practicing. The purpose and methods of the study, freedom of participation and discontinuation, and protection of privacy were fully explained to the participants, and written informed consent was obtained. This study was approved by the Ethical Review Committee of the International University of Health and Welfare (18-Io-21).

During the 2-month study period, 3 young and 12 older adults dropped out of the study because of either a busy schedule, poor physical condition during the summer, or the start of other physical activities. Finally, 35 young adults (age: 27.5 ± 3.7 years, height: 164.5 ± 10.3 cm, weight: 59.7 ± 13.5 kg, body mass index: 21.8 ± 2.9, 19 females) and 29 older adults (age: 72.7 ± 5 years, height: 154.5 ± 6.9 cm, weight: 55.9 ± 9.1 kg, body mass index: 23.4 ± 3.2, 26 females) completed the study. All participants completed the study without any adverse events.

### 2.2. Study Design and Implementation Procedure

The study flowchart is shown in Figure 1. This study included a pre–post-intervention design; measurements were taken three times: at baseline/pre-intervention (before), post-intervention (after 1 month), and at follow-up (after 2 months). DS exercises were conducted for the first month, with weekly follow-ups via phone or e-mail. In the second month, only measurements were taken, and no intervention was conducted. To assess the full range of physical abilities, lower limb muscle strength, static balance, dynamic balance, agility, and flexibility were measured. In addition, a questionnaire was administered as a subjective evaluation to determine whether the DS exercises were easily executed and sustained.

#### 2.2.1. DS Exercises

DS exercises were performed barefoot on a non-slip floor with no objects nearby to reduce the risk of injury in case of falls. The length of each side of the diamond shape used in the DS exercises was half the participant’s height, with an acute angle of 60° and an obtuse angle of 120° (Figure 2). The procedure to create the diamond shape was taught to the participants as follows. (1) Two strings (long and short, diagonal of the diamond) individualized for each participant’s height were given. (2) After checking the practice space, the two strings were arranged so that the midpoints of the two strings intersected perpendicularly. (3) The four end points of the strings became the vertices of the DS, and markers were placed at each vertex (orange dots in Figure 2).

Participants were instructed to walk according to the procedure shown in Figure 2 (no crossing of the feet) and to keep the trunk facing forward at all times. Each participant was instructed to walk at an optimal speed and to change freely between right and left turns. The practice time was set to 3 min using a timer.

A “calendar for DS exercise” form was distributed to the participants to check their exercise performance. The calendar form was filled out with the date and day of the week when practice was conducted. Additionally, the practice progress was checked telephonically or by e-mail once a week during the practice period. The DS exercises were performed continuously for 3 min, four times a week, for 1 month. Participants were instructed to continue with their existing lifestyle, with the exception of including DS exercises (no additional exercises were to begin and no preexisting exercises were to cease) for the duration of the study.

#### 2.2.2. Assessment of Physical Performance

Physical performance was assessed using the following eight-item sequence in a randomized manner. In all assessment procedures, a skilled physical therapist explained the methods and procedures before taking the measurements, and the participant performed the procedures barefoot.

#### 2.2.3. Lower Limb Muscle Strength

To evaluate the muscle strength of the lower limbs, we employed the 30 s chair stand test (CS-30), which does not require a measuring device and is commonly used in clinical practice. The CS-30 measures the number of times a person can stand up in 30 s from a chair approximately 40 cm in height [23]. If the 30 s ended while the participant was in the process of standing up, this was counted as one time if the torso and knees were already extended. Only one measurement was taken.

#### 2.2.4. Static Balance

The functional reach test (FRT) and standing on one leg with eyes closed (SOLEC) test were performed to evaluate static balance. The FRT measures the reaching distance of the right upper limb with maximum forward reach. Three tests were performed, and the maximum value was taken [22]. The SOLEC test measures the length of time an individual can stand on one leg (dominant leg) with their eyes closed. Two tests were conducted, and the maximum value was taken [22].

#### 2.2.5. Dynamic Balance

To evaluate dynamic balance ability, the time required for five rounds of DS (5-DS), left-right DS (LRDS), and the modified Y balance test (YBT) was measured.

For 5-DS, after practicing three laps of the diamond shape shown in Figure 2, the time taken to walk five laps at the fastest possible speed from the dominant foot side was measured twice, and the average value was taken [22].

For LRDS, each participant practiced three laps of the diamond shape shown in Figure 2 and then walked around the diamond twice at maximum speed, the first time beginning with the dominant foot and the second time beginning with the non-dominant foot. The times were then averaged [22].

The YBT was performed with only 135° posterolateral lower limb reaching exercises (YBT-L; YBT-R) bilaterally [24]. The reason for this was to accommodate the older participants in this study. As shown in previous studies, the YBT is intended for younger people who play sports [25,26]. Therefore, to reduce the burden on the older participants, only a portion of the YBT was employed. The supporting foot stayed on the floor, and the toes of the reaching foot were allowed to touch as far as possible. The maximum reach distance that could be achieved was then measured. Six trials were performed, including the exercises, and the maximum value was taken.

#### 2.2.6. Agility

To evaluate agility, we performed the open–close stepping test (OCS-10). The OCS-10 measures the number of times the participant can open and close both legs simultaneously in a chair while in a sitting position in 10 s. After one exercise, the number was measured twice at an interval of approximately 30 s, and the maximum value was taken [22].

#### 2.2.7. Flexibility

The finger-to-floor distance (FFD) was used to evaluate trunk flexibility [27]. The FFD was measured from the tip of the third finger to the floor by having the participant stand on a 30 cm platform and bend the body forward and downward (knee extension). After one practice session, two measurements were taken at 20 s intervals, and the maximum value was taken.

### 2.3. Subjective Evaluation of DS Exercises

Training in a rehabilitation setting is often long-term and, in some cases, must be continued after hospital discharge. Therapists need to devise training methods that are easy to continue, and the participants’ motivation and subjective evaluation of the training should also be considered. Motivation for rehabilitation has been shown to play a dominant role in the rehabilitation process and promotes a reduction in the rate of disability [28]. Therefore, a subjective evaluation was conducted to establish whether the DS exercises were easy to adhere to for healthy participants. The subjective ratings were difficulty, achievement, and lightness/enjoyment in performing the DS exercises. Questionnaires were administered on a 5-point scale: not at all, a little, somewhat, very much, and greatly.

The evaluation was conducted twice––after the first practice (after) and at 1 month (after 1 month). Difficulty was defined as how difficult it was for the participants to perform the DS exercises as instructed and to practice for 3 min, four times a week, for a month. Achievement refers to the degree to which the participants were able to accomplish what they were instructed to do. Lightness/enjoyment refers to the degree to which the participants felt light and happy when they performed the DS exercises.

### 2.4. Statistical Analysis

The sample size was determined using G Power version 3.1.9.7 (Universität Kiel, Kiel, Germany) based on effect size f of 0.25, α error probability at 0.05, and power (1-β error probability) at 0.90. A minimum sample size of 18 subjects per group was required; due to the long study duration and the lifestyle requirements of the participants, we anticipated many dropouts. Therefore, we set a larger dropout rate of 30% to 50% and recruited approximately twice as many participants per group.

The Kolmogorov–Smirnov test confirmed the normality of most measurement results. Two-way analysis of variance was employed to determine whether there were significant differences related to age (young and older adults) and time (before, after 1 month, and after 2 months) in the assessment of physical performance. The effect size was evaluated using *η_p_^2^*. When an interaction was found within each parameter, multiple comparisons (Bonferroni) of the within-participant factors (before, after 1 month, and after 2 months) were performed.

The subjective evaluation was analyzed by employing cross-tabulations and chi-square tests to determine the pattern of change in impressions (after and after 1 month) after the exercises. Statistical analyses were performed using SPSS^®^ statistical software version 25 (IBM Corporation, Armonk, NY, USA). Statistical significance was set at *p* < 0.05.

## 3. Results

The results for all physical performance measures are presented in Table 1. Two-way analysis of variance showed a main effect of age and time for all items and an interaction effect for four items (Table 2). Details of the items for which interaction effects were found are shown in Figure 3.

The results of the subjective evaluation are presented in Table 3. Only the level of difficulty among older adults showed a significant decrease (*p* = 0.001). No significant differences were found for the items of achievement and lightness/enjoyment.

## 4. Discussion

This study investigated the effects of DS exercises on improving balance in older and young adults. Significant interactions were found among SOLEC, LRDS, YBT-L, and YBT-R. Among them, LRDS showed a significant decrease after 1 and 2 months of intervention compared with the pre-intervention period in both young and older adults. Furthermore, LRDS did not change significantly between 1 and 2 months post-intervention. In the observation period after the end of practice, LRDS did not increase without practice, suggesting that there may have been a maintenance effect from practice. As a component of balance ability, LRDS includes static stability, motor function, stability limits, predictive postural control, dynamic stability, and sensory function [22]. Therefore, DS exercises comprehensively improved the components of balance ability, regardless of age, suggesting that the effect may continue after the training intervention. However, dynamic balance evaluation using LRDS is measured using movements similar to the DS exercises. It is possible that a learning effect of the repetition of similar movements was obtained.

In the young group, SOLEC showed a significant decrease after 1 and 2 months of intervention compared with the pre-intervention period. Furthermore, SOLEC did not change significantly after 1 and 2 months of intervention. This finding is consistent with the results obtained regarding LRDS, suggesting that DS exercises improve multiple components of balance ability in the young group and may have a lasting effect. In contrast, no significant differences were noted for SOLEC in the older adult group before, 1 month after, or 2 months after the intervention. SOLEC includes three balance components: sensory integration, static stability, and motor function. Sensory integration, static stability, and motor function in older adults have all been reported to decline with age [29]. In particular, older adults’ dependence on visual information included in sensory integration may be a factor in the low value of approximately 3 s. Therefore, it is possible that SOLEC was not significantly different before and after the intervention because of its high difficulty for older adults.

Both YBT-R and YBT-L showed significant increases after 1 and 2 months of intervention compared with the pre-intervention period in young and older adults. Similar to the results obtained regarding LRDS, these results suggest that YBT did not decrease even without exercise during the observation period after the completion of the exercises and that there may have been a maintenance efficacy of the exercises. It is possible that the specialized DS exercises enhanced backward oblique balance, since backward oblique movements are less common in daily life movements and exercises.

Furthermore, although this study focused on the balance element to explore the effects of DS exercise, the positive effect on lower extremity muscle strength and function should not be overlooked. Recent research reports indicate that the decline in muscle quality and function with aging has a dramatic impact on autonomy and quality of life, and is a serious concern [3].

Subjective feelings also have a significant impact on quality of life. Older adults showed a significant decrease in the level of difficulty of DS exercises after 1 month of the exercise intervention compared with that after the initial exercises. Learning to move the body smoothly occurs through repetitive practice [30]. As a result, the level of difficulty may have decreased as proficiency in DS movements increased. The subjective ratings of achievement and lightness/enjoyment showed no significant changes in either group, and positive results were obtained both after the initial exercises and 1 month after the exercise intervention. These items were at high levels from the beginning, indicating that participants may have enjoyed the exercises and continued to practice. Long-term exercise continuation requires intrinsic motivation through self-efficacy and enjoyment, and increased motivation results in positive changes at physical, psychological, and social levels [31,32,33]. Therefore, we believe that the DS exercises in this study not only improved balance and lower limb muscle strength in the older participants, but may have also had a positive impact on their quality of life.

This study has some limitations. First, we evaluated specific physical functions and did not assess the overall balance ability. Future studies should assess balance ability using assessment methods such as the Berg Balance Scale or Balance Evaluation Systems Test. Second, this study was not blinded to the intervention or evaluation. Future studies should include study designs such that the person teaching the exercises and the evaluator are different people, with the evaluator blinded to the intervention. Third, all outcome measures used in this study did not take into account the psychological characteristics of the participants. Future studies require a research design that can take into account the psychological characteristics of the participants’ willingness to work hard.

This study suggests that DS exercises improve balance ability in both young and older adults and suggests a sustained effect. Furthermore, DS exercises have the potential to be an intervention method that is appropriate in terms of movement difficulty and is easy to sustain. These exercises may be a means of effectively improving balance ability, regardless of age. Because stepping movements in the left and right backward diagonal directions are infrequent in daily life, it is likely that older adults in particular are prone to backward diagonal balance loss. The DS exercises in this study are likely to be useful in improving balance in the back oblique direction because they are specialized for these movements. Minimizing the number of trials (minimum repetitions) has been reported to increase learning effectiveness due to simplicity and ease to apply [34]. The DS exercises performed in this study have the potential to improve balance in a short period of time, with only 3 min per session, four times per week, for 1 month. Therefore, the practical significance of this study is that DS exercises are likely to effectively improve physical performance, especially in the backward oblique direction, which is prone to decline, and to improve balance, even if DS exercises are used during small gaps in daily life (short duration). Exercise is considered a panacea that keeps older participants energetic and improves age-related functional decline [35]. Therefore, it is considered very important for training to be enjoyable and easily sustainable. Approximately one in three community-dwelling older adults fall each year, according to a systematic review from 2019 [36]. This systematic review summarized 108 randomized controlled trials (RCTs) of 23407 community-dwelling older participants in 25 countries. The main findings were that exercise (all types) reduced fall rates by 23% compared to a control group (those not thought to reduce falls). Exercise also reduced the number of people who had experienced one or more falls by 15%. Therefore, DS exercises that include a variety of balance elements are more likely to be an effective fall prevention exercise for older adults. Based on the results of this study, DS exercises could easily be generalized as training to improve balance and lower limb muscle function in older participants who are walking independently.

Additionally, balance exercises to improve postural control and stability are effective in preventing recurrent ankle sprains and CAI [19,20,37]. A sensory function-oriented approach is important in the rehabilitation of postural instability in patients with CAI [38]. The Diamond Steps Test includes elements of sensory function, dynamic stability, and predictive and reactive postural control [22]; we believe that DS exercises may help prevent the recurrence of ankle sprains and CAI. Therefore, for further development of our research, we would like to study the effect and course of DS exercises in preventing the recurrence of ankle sprains and CAI in the future in a sports population with a history of ankle sprains and CAI.

## 5. Conclusions

DS exercises are likely to enhance balance (especially in the backward oblique direction) in a short period of time. The training is sufficiently challenging and easy to continue independently for older participants.

## Figures and Tables

**Figure 1 healthcare-11-01834-f001:**
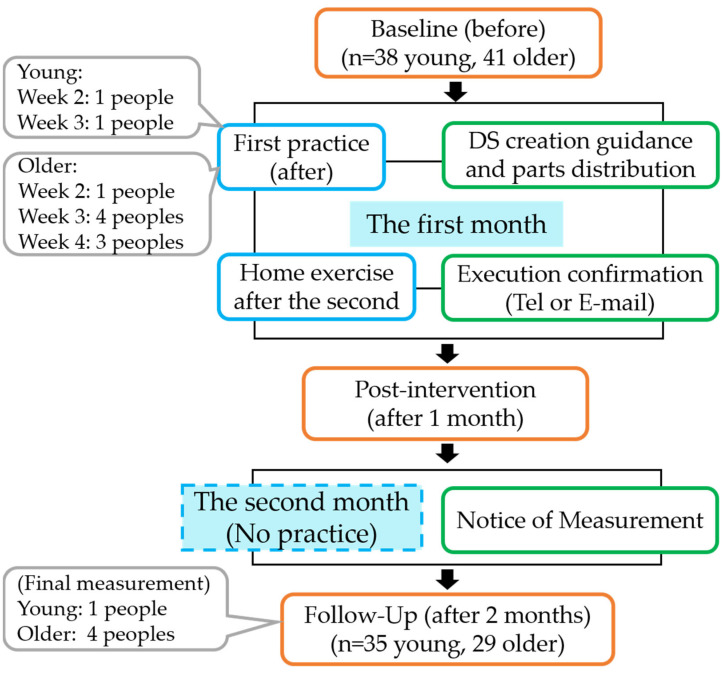
Flowchart of this study. Key: gray frame, dropouts; blue frame, training; orange frame, measurement; green frame, other.

**Figure 2 healthcare-11-01834-f002:**
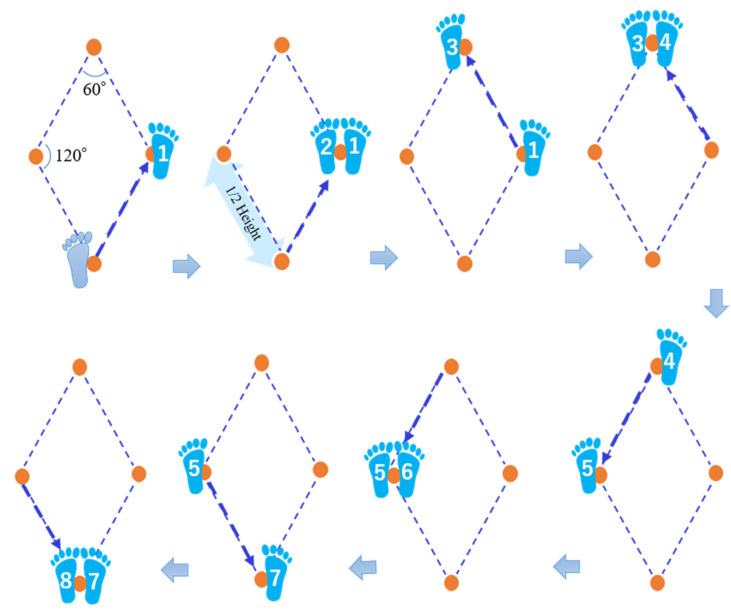
Diamond steps exercises. The numbers on the feet indicate the order of steps. The arrow indicates the direction of travel.

**Figure 3 healthcare-11-01834-f003:**
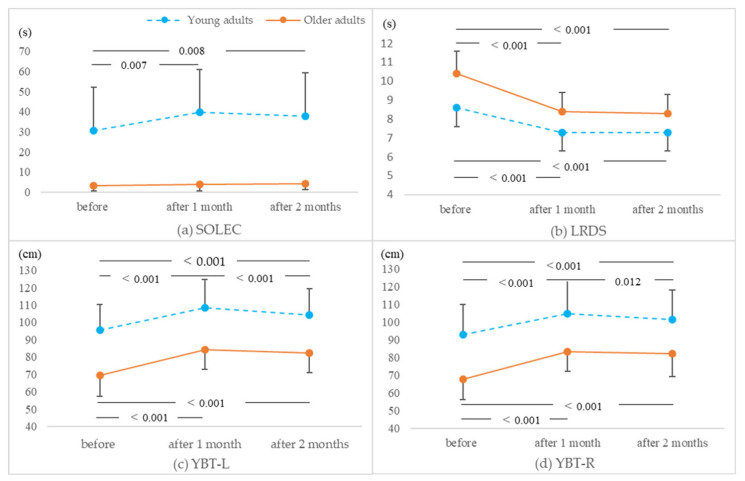
Changes over time in items for which interactions were observed. Abbreviations: SOLEC, standing on one leg with eyes closed; LRDS, left–right diamond steps; YBT, Y balance test; YBT-L, YBT was performed with a 135° posterolateral lower limb reaching motion of the left foot; YBT-R, YBT was performed with a 135° posterolateral lower limb reaching motion of the right foot.

**Table 1 healthcare-11-01834-t001:** Results of balance measurement items (*n* = 64).

Test (Units)	Young Adults			Older Adults		
	Before	After 1 Month	After 2 Months	Before	After 1 Month	After 2 Months
CS-30 (no. of times)	25.5 ± 6.2	30.9 ± 6.3	30.5 ± 6.5	19.3 ± 6.7	25.2 ± 7.9	25.1 ± 6.7
FRT (cm)	38.8 ± 5.6	41.5 ± 4.7	40.7 ± 4.5	28.5 ± 5.9	32.5 ± 4.7	31.2 ± 4.9
SOLEC (s)	30.5 ± 21.6	39.8 ± 21.1	37.8 ± 21.4	3.2 ± 2.5	4.0 ± 3.3	4.1 ± 2.9
5-DS (s)	23.3 ± 3.7	17.7 ± 2.4	18.0 ± 2.7	26.9 ± 6.5	20.5 ± 3.5	21.2 ± 3.7
LRDS (s)	8.6 ± 1.2	7.3 ± 1.0	7.3 ± 1.0	10.4 ± 2.3	8.4 ± 1.6	8.3 ± 1.4
YBT-L (cm)	95.6 ± 14.9	108.6 ± 16.3	104.1 ± 15.2	69.3 ± 11.9	84.2 ± 11.5	82.3 ± 11.4
YBT-R (cm)	92.8 ± 17.2	104.8 ± 18.4	101.5 ± 16.5	67.7 ± 11.3	83.4 ± 11.3	82.1 ± 12.8
OCS-10 (no. of times)	16.6 ± 2.8	19.4 ± 2.7	19.4 ± 2.7	13.6 ± 2.0	16.2 ± 2.4	16.1 ± 2.5
FFD (cm)	2.1 ± 9.5	5.3 ± 9.1	3.9 ± 9.3	5.9 ± 7.9	9.5 ± 7.0	9.0 ± 7.5

Data are presented as mean ± SD. Abbreviations: CS-30, 30 s chair stand test; FRT, functional reach test; SOLEC, standing on one leg with eyes closed; 5-DS, time required for five rounds of diamond steps; LRDS, left–right diamond steps; YBT, Y balance test; YBT-L, YBT was performed with a 135° posterolateral lower limb reaching motion of the left foot; YBT-R, YBT was performed with a 135° posterolateral lower limb reaching motion of the right foot; OCS-10, open–close stepping test; FFD, finger-to-floor distance.

**Table 2 healthcare-11-01834-t002:** Results of two-way analysis of variance (ANOVA) (*n* = 64).

Test (Units)	Age Group			Time Lapse			Age Group × Time Lapse		
	F-Value	*p*-Value	η_p_^2^	F-Value	*p*-Value	η_p_^2^	F-Value	*p*-Value	η_p_^2^
CS-30 (no. of times)	12.613	0.001	0.169	114.838	<0.001	0.649	0.435	0.648	0.007
FRT (cm)	68.921	<0.001	0.526	29.231	<0.001	0.320	1.137	0.324	0.018
SOLEC (s)	76.571	<0.001	0.553	7.677	0.001	0.110	5.045	0.008	0.075
5-DS (s)	13.623	<0.001	0.180	141.244	<0.001	0.695	0.430	0.652	0.007
LRDS (s)	15.324	<0.001	0.198	109.602	<0.001	0.639	4.762	0.010	0.071
YBT-L (cm)	52.795	<0.001	0.460	133.568	<0.001	0.683	3.127	0.047	0.048
YBT-R (cm)	36.772	<0.001	0.372	108.966	<0.001	0.637	4.180	0.018	0.063
OCS-10 (no. of times)	27.560	<0.001	0.308	125.835	<0.001	0.670	0.234	0.792	0.004
FFD (cm)	4.333	0.042	0.065	62.433	<0.001	0.502	2.373	0.097	0.037

Abbreviations: ANOVA, analysis of variance; Age Group, young and older adults; Time Lapse, before, after 1 month, after 2 months; CS-30, 30 s chair stand test; FRT, functional reach test; SOLEC, standing on one leg with eyes closed; 5-DS, time required for five rounds of diamond steps; LRDS, left–right diamond steps; YBT, Y balance test; YBT-L, YBT was performed with a 135° posterolateral lower limb reaching motion of the left foot; YBT-R, YBT was performed with a 135° posterolateral lower limb reaching motion of the right foot; OCS-10, open–close stepping test; FFD, finger-to-floor distance.

**Table 3 healthcare-11-01834-t003:** Results of subjective evaluation (*n* = 64).

		Not at All	A Little	Somewhat	Very Much	Greatly	Cramer’s V	*p*-Value
Young adults (*n* = 35)								
Difficulty	After	7 (20%)	15 (43%)	9 (26%)	4 (11%)	0 (0%)	0.236	0.274
	After 1 month	13 (37%)	14 (40%)	7 (20%)	1 (3%)	0 (0%)		
Lightness and enjoyment	After	0 (0%)	3 (9%)	12 (34%)	19 (54%)	1 (3%)	0.250	0.223
	After 1 month	0 (0%)	2 (6%)	6 (17%)	23 (66%)	4 (11%)		
Sense of accomplishment	After	0 (0%)	10 (29%)	13 (37%)	11 (31%)	1 (3%)	0.234	0.428
	After 1 month	1 (3%)	7 (20%)	16 (45%)	9 (26%)	2 (6%)		
Older adults (*n* = 29)								
Difficulty	After	3 (10%)	15 (52%)	10 (35%)	1 (3%)	0 (0%)	0.544	0.001
	After 1 month	18 (62%)	6 (21%)	4 (14%)	1 (3%)	0 (0%)		
Lightness and enjoyment	After	0 (0%)	0 (0%)	20 (69%)	9 (31%)	0 (0%)	0.278	0.213
	After 1 month	0 (0%)	1 (3%)	14 (48%)	12 (41%)	2 (7%)		
Sense of accomplishment	After	0 (0%)	2 (7%)	7 (24%)	14 (48%)	6 (21%)	0.153	0.852
	After 1 month	1 (3%)	3 (10%)	7 (24%)	12 (41%)	6 (21%)		

Values are expressed as number of people (%); Cramer’s V: effect size.

## Data Availability

The data that support the findings of this study are available from the corresponding author (S.S.) upon reasonable request. Written informed consent has been obtained from all participants to publish this paper.

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
