# Peer review of "Effects of Diamond Steps Exercises on Balance Improvement in Healthy Young and Older Adults: A Protocol Proposalâ€"

_healthcare, 2023, doi:10.3390/healthcare11131834_

Round 1
Reviewer 1 Report
Title
As it appears to me, the protocol was developed by the authors. If so, it should appear in the title: a protocol proposal.
Introduction
This sentence is out of context: Therefore, both older and young adults should adopt measures to maintain and improve balance. The introduction had been describing ankle sprain numbers and suddenly the phrase came up.
Materials and Methods
Participants
Was it a convenience sample? If not, state how the sample size was calculated.
DS exercises
Was the protocol developed by the authors? If not, there must be a reference.
Static balance
The FRT evaluates static balance only in the anterior direction and if the participant has pain in the lumbar spine or a decrease in the flexibility of the lumbar spine, the results will be affected, not necessarily meaning change in balance.
Which leg was the SOLEC test done? Dominant? It needs to be described.
Statistical analyses (the english is wrong)
To perform an analysis of variance there are a number of prerequisites. Make it clear that the ANOVA was done after this check and that all prerequisites were met.
Results
It is necessary to insert a flowchart for experimental studies.
There is repetition of data in the results section (text and figures, text and tables) making reading tiring.
Discussion
There is a repetition of results in the discussion section. Authors should only discuss the results and not show them again.
To change balance test values, it is necessary to have activities that moderately or highly challenge the systems that collaborate to maintain postural balance. It is necessary to discuss how the Diamond Steps Exercises do this.
Conclusions
The conclusions are from the study and, therefore, there should be no bibliographic references in this section.
"Furthermore, DS exercises have the potential to be an intervention method that is appropriate in terms of movement difficulty and easy to sustain. These exercises may be a means of effectively improving balance ability, regardless of age. Because stepping movements in the left and right backward diagonal directions are infrequent in daily life, it is likely that older adults, in particular, are prone to backward diagonal balance loss. The DS exercises in this study are likely to be useful in improving balance in the back oblique direction because they are specialized for these movements. Minimizing the number of trials (minimum repetitions) has been reported to increase learning effectiveness because it is simple and easy to apply [26]." This is not a conclusion. This should be inserted in the discussion section.
"The study suggest that DS exercises improve balance ability in both young and older adults, suggesting a sustained effect."
"The DS exercises performed in this study have the potential to improve balance in a short period of time, with only 3 min per session, four times per week, for 1 month."
Authors must unite the two sentences, writing a single conclusion.
References
73% of the references have more than 5 years of publication. Need to update
Author Response
Response to Reviewer 1 Comments
All revised parts of the paper are shown in green font.
Title
Comment: As it appears to me, the protocol was developed by the authors. If so, it should appear in the title: a protocol proposal.
Response:
Thank you for pointing this observation and suggestion. We have revised the title accordingly. (Page 1, Line 3)
Introduction
Comment: This sentence is out of context: Therefore, both older and young adults should adopt measures to maintain and improve balance. The introduction had been describing ankle sprain numbers and suddenly the phrase came up.
Response: Thank you for this observation. We apologize for the illogical flow in the Introduction. We attempted to reduce the word count, which impacted the paragraph flow of the Introduction. We have made major revisions to the Introduction and would appreciate your review. (Page 1-2, Lines 34-72)
Materials and Methods
Participants
Comment: Was it a convenience sample? If not, state how the sample size was calculated.
Response: Thank you for your question and suggestion. We have added the sample size calculation method to "2.4. Statistical analyses". We would appreciate it if you could check it. (Page 6, Lines 236-241)
DS exercises
Comment: Was the protocol developed by the authors? If not, there must be a reference.
Response: As you have speculated, the Diamond Steps Test was devised by the author. We have corrected the text accordingly. (Page 2, Lines 70-72)
Static balance
Comment: The FRT evaluates static balance only in the anterior direction and if the participant has pain in the lumbar spine or a decrease in the flexibility of the lumbar spine, the results will be affected, not necessarily meaning change in balance.
Response: Thank you for this correct observation. For that reason, the exclusion criteria for the subject recruitment was "...those with pain...". We have revised the text accordingly.
(Page 2, Lines 82-85)
Comment: Which leg was the SOLEC test done? Dominant? It needs to be described.
Response: Thanks for pointing this out. The test was performed with the dominant foot. The text has been revised accordingly. (Page 5, Line 182)
Comment: Statistical analyses (the english is wrong)
Response: Thank you for pointing this out. We have corrected the text accordingly. (Page 6, Line 235)
Comment: To perform an analysis of variance there are a number of prerequisites. Make it clear that the ANOVA was done after this check and that all prerequisites were met.
Response: Thank you for this observation and suggestion. The statistics requirements were checked again and we confirmed that the conditions for the application of analysis of variance were met. The conditions for application of analysis of variance are as follows: (1) The Kolmogorov-Smirnov test showed the normality of most of the measurement results; (2) The nature of the data were ratio and interval scales; (3) The data are meaningful to compare means; (4) The data in this study were measured with two groups and three factors. The first condition was added to the text [2.4. Statistical analysis]. (Page 7, Lines 242-243)
Results
Comment: It is necessary to insert a flowchart for experimental studies.
Response: Thank you for your suggestion. A flowchart of the study (Figure 1) has been added.
(Page 3, Line 106, Lines 115-119)
Comment: There is repetition of data in the results section (text and figures, text and tables) making reading tiring.
Response: Thank you for pointing this out. In an attempt to briefly summarize the results, I have revised the results to be more concise. (Page 6, Lines 256-259)
Discussion
Comment: There is a repetition of results in the discussion section. Authors should only discuss the results and not show them again.
Response: Thank you for pointing that out. To limit confusion regarding the topics of discussion, we have presented each item to discuss each one separately. Also, considering that Healthcare has many readers who are not rehabilitation specialists, we considered it appropriate to elaborate on each item. Therefore, it would be very helpful if you could allow grace in this section for better clarity to the readers.
Comment: To change balance test values, it is necessary to have activities that moderately or highly challenge the systems that collaborate to maintain postural balance. It is necessary to discuss how the Diamond Steps Exercises do this.
Response: Thank you for your comment and suggestion.
The DS exercise was used for the Diamond Steps Test, which was designed based on the balance system theory. The Diamond Steps Test is a test of the seven elements of (1) static stability, (2) underlying motor systems, (3) functional stability limits, (5) reactive postural control, (6) anticipatory postural control, (7) dynamic stability, and (8) sensory integration. Two elements, namely ,(4) verticality and (9) cognitive influences were not verified in this test. Therefore, we hypothesized that the Diamond Steps used in the Diamond Steps Test could be used as a practice method to increase the activity of collaborative systems that maintain postural balance. This was discussed in the Introduction section and further elaborated in the Discussion.
(Page 2, Lines 64-72)
Conclusions
Comment: The conclusions are from the study and, therefore, there should be no bibliographic references in this section.
"Furthermore, DS exercises have the potential to be an intervention method that is appropriate in terms of movement difficulty and easy to sustain. These exercises may be a means of effectively improving balance ability, regardless of age. Because stepping movements in the left and right backward diagonal directions are infrequent in daily life, it is likely that older adults, in particular, are prone to backward diagonal balance loss. The DS exercises in this study are likely to be useful in improving balance in the back oblique direction because they are specialized for these movements. Minimizing the number of trials (minimum repetitions) has been reported to increase learning effectiveness because it is simple and easy to apply [26]." This is not a conclusion. This should be inserted in the discussion section.
"The study suggest that DS exercises improve balance ability in both young and older adults, suggesting a sustained effect."
"The DS exercises performed in this study have the potential to improve balance in a short period of time, with only 3 min per session, four times per week, for 1 month."
Authors must unite the two sentences, writing a single conclusion.
Response: Thank you for your remarks and suggestions. We have revised this section and kindly request your approval of the revision. (Page 11, Lines 389-391)
References
Comment: 73% of the references have more than 5 years of publication. Need to update
Response: Thank you very much for your suggestion. There were a few references that remained necessary, however I have updated the majority. Thank you very much.
(Page 11-13, Lines 422-509)

Reviewer 2 Report
The authors write that the aim of the manuscript entitled ‘Effects of Diamond Steps Exercises on Balance Improvement in Healthy Young and Older Adults: An Experimental Study’ was to verify whether Diamond Steps exercises contribute to the improvement of balance ability in young and older adults and analyse whether these exercises are easy to continue in these participants. I read this manuscript with great interest. The manuscript is well written, but I suggest some corrections to improve quality of its final version.
Detailed comments and suggestions:
Abstract is well written. However, I suggest correcting the number of people who participated in the study to the number of people who completed the study and whose results were used in the final analyzes (35 young adults; 29 older adults).
The introduction is very concise but contains enough information to justify the choice of research topic presented in this manuscript.
Materials and Methods:
- lines 69-76: section 2.1. Participants should be supplemented with information on the number of people who completed participation in the study and whose results were used for final analyses. Since 38 young adults and 41 older adults were included in the study, and the conclusions were formulated based on the results obtained by 35 young adults and 29 older adults, this should be specified here. I suggest moving the text from lines 194-198 to this section, or placing a flow chart here, containing relevant information about the number of subjects who did not complete the study.
- lines 138-152: the authors state that dynamic balance was assessed, among others, with the Y-balance Test (line 139) in accordance with the criteria of the Star excursion balance test presented by Plisky et al. (lines 148-149). Why, then, only one of the three recommended directions of reaching the leg was performed and the normalization of the results considering the length of the lower limb was not taken into account. Plisky et al. in the cited article describes detailed recommendations for performing and interpreting the results of the Y-balance test. The results of later studies have shown that the most measurable direction of leg reach in this test is the anterior direction. Since the authors performed the test only in the external posterior direction, this choice should be clarified. I suggest the authors consider supplementing the name of the test used with 'modified Y-balance test'.
- line 148: the authors should correct the typo in the test name from 'the Stat excursion balance test' to 'Star excursion balance test'.
- lines 155-159: why there is no reference to scientific publications describing the Open Close Stepping Test (OCS-10)? It is worth supplementing this by referring to other authors who have described or used this test.
- Line 162: Why do the authors refer to the 1975 publication when describing the test used to assess trunk flexibility? Please replace this reference and cite authors who published their research at least after 2010.
Results:
A detailed presentation of the results allows the reader to understand each step made by the authors, leading to the final conclusions. However, I suggest making the following adjustments:
- lines 194-198: I suggest moving this paragraph as it stands to section 2.1. Participants.
- Table 2: I suggest moving the explanation of the ANOVA abbreviation from the footer below the table to the table title. The table may be titled 'The results of two-way analysis of variance (ANOVA)’. In the footer below the table, only the abbreviations included in the table body should be explained.
Discussion and Conclusions:
I have no comments for these sections.
Reviewer 3 Report
Dear authors,
Firstly, I would like to congratulate you on the work done. This study aims to evaluate the effect of a short intervention program on balance in adults and in the elderly. Regarding the manuscript, in my humble opinion and although some flaws that should be addressed, I found it suitable for publication.
Below you may find some remarks / suggestions on your work:
The introduction is too short and does not adequately address the issue you are trying to study, nor it justifies the relevance of your work. It needs to be expanded and improved. Also, the connection with the research field is not well done, I suggest you rewrite this section with information more in line with the aims and goals of your work.
The methods are generally well written, just minor issues:
Did you check if the participants were active or sedentary? Don’t you think this may influence the impact of your intervention?
I would suggest you add references for all your tests, as some of them have and other don’t.
Line 147 you write posterior outward. Why not posterolateral?
Line 148 correct “Stat”. I think you mean “Star”
Did you normalize the YBT results for lower limb size?
Regarding the subjective evaluation of the DS exercises, please provide information regarding the validation process of this questionnaire. Has it been used before, or was it a new one?
Regarding the statistical analysis, please provide G*power estimation for this sample.
Regarding the results, they are well presented and easy to follow. I would only suggest to change the position of figures 2, as they are right after a table and without any text in between.
The discussion seems appropriate. Please add practical implications of your work
The conclusions are correctly written according to your results
Once again, congratulations on the job well done, I hope my remarks help you improve it.
Quality of English seems fine, just proofread.
Reviewer 4 Report
“Effects of Diamond Steps Exercises on Balance Improvement in Healthy Young and Older Adults: An Experimental Study.” It is a novel study with convincible background, methodology, results, and discussion. However, the authors are requested to address the following recommendations.

Minor English correction required
Reviewer 5 Report
I thank the authors for the work done. I proceed to the suggestions step by step:
Abstract
- I would suggest the authors to re-write the abstract, using a classical structure, and inserting the results. This ultimately gives visibility to the manuscript if published.
- Tip: Keywords must be different from the words in the title to optimize the search for the manuscript through the search engines.
The general rule: The search engine algorithm works by searching for keywords present in the first 65 characters of the manuscript title. Then it searches for the keywords present in the first two sentences of your abstract and finally searches for keywords other than the title words.
Introduction
The introduction is well written, however, a recent review has analyzed the decline in muscle mass and function that represents one of the most problematic changes associated with aging, and has dramatic effects on autonomy and quality of life. This manuscript could strengthen the introduction by explaining the relationship between sarcopenia and falls in the elderly.
Agostini D, et al. An Integrated Approach to Skeletal Muscle Health in Aging. Nutrients. 2023 Apr 7;15(8):1802. doi: 10.3390/nu15081802. PMID: 37111021; PMCID: PMC10141535.
Methods
Better explain recruitment procedures
The explanation of DS Exercise is short and unclear. Expand
Participants
-How did you decide on the sample size before starting the study? Have you carried out an a priori power analysis? For example, with G*Power
The results are very clear
Discussion
The discussions are clear and to the point.
The authors' conclusions are justified. The take-home message is clear.
Round 2
Reviewer 1 Report
The authors adhered to most of the suggested changes, making the manuscript more pleasant to read.
Reviewer 3 Report
Congratulations on the improvements.
I believe that the manuscript has improved substantially. However, I still think that the introduction could better enphasize the relevance of the research, particularly regarding the way you include references about injuries in athletes (lines 53-55). Why not include something regarding the elderly?
Nonetheless, I am satisfied with the revisions made. I believe that the work is worth publishing.
Congratulations
I am satisfied with the quality of English.
Reviewer 4 Report
Minor English editing required
Dear Editor
Manuscript can be accepted after minor English corrections